# Detection of SARS-CoV-2 in Wastewater Associated with Scientific Stations in Antarctica and Possible Risk for Wildlife

**DOI:** 10.3390/microorganisms12040743

**Published:** 2024-04-06

**Authors:** Marcelo González-Aravena, Cristóbal Galbán-Malagón, Eduardo Castro-Nallar, Gonzalo P. Barriga, Víctor Neira, Lucas Krüger, Aiko D. Adell, Jorge Olivares-Pacheco

**Affiliations:** 1Departamento Científico, Instituto Antártico Chileno, Punta Arenas 6200985, Chile; lkruger@inach.cl; 2GEMA, Center for Genomics, Ecology & Environment, Universidad Mayor, Santiago 8580745, Chile; cristobal.galban@mayor.cl; 3Anillo en Ciencia y Tecnología Antártica POLARIX, Santiago 8370146, Chile; ecastron@utalca.cl; 4Institute for Environment, Florida International University, Miami, FL 33199, USA; 5Departamento de Microbiología, Facultad de Ciencias de la Salud, Universidad de Talca, Campus Talca, Talca 3481118, Chile; 6Centro de Ecología Integrativa, Universidad de Talca, Campus Talca, Talca 3460000, Chile; 7Laboratorio de Virus Emergentes, Programa de Virología, Instituto de Ciencias Biomédicas, Facultad de Medicina, Universidad de Chile, Santiago 8380453, Chile; gonzalo.barriga@uchile.cl; 8Medicina Preventiva Animal, Facultad de Ciencias Veterinarias, Universidad de Chile, Santiago 8820808, Chile; victorneira@u.uchile.cl; 9Millennium Institute Biodiversity of Antarctic and Subantarctic Ecosystems (BASE), Santiago 7750000, Chile; 10Escuela de Medicina Veterinaria, Facultad de Ciencias de la Vida, Universidad Andrés Bello, Santiago 9350841, Chile; aiko.adell@unab.cl; 11Millennium Initiative for Collaborative Research on Bacterial Resistance, MICROB-R, Santiago 7550000, Chile; 12Grupo de Resistencia Antimicrobiana en Bacterias Patógenas y Ambientales, GRABPA, Instituto de Biología, Pontificia Universidad Católica de Valparaíso, Valparaíso 2373223, Chile

**Keywords:** Antarctica, environmental surveillance, wastewater-based epidemiology, SARS-CoV-2, COVID-19, Antarctic wildlife

## Abstract

Before December 2020, Antarctica had remained free of COVID-19 cases. The main concern during the pandemic was the limited health facilities available at Antarctic stations to deal with the disease as well as the potential impact of SARS-CoV-2 on Antarctic wildlife through reverse zoonosis. In December 2020, 60 cases emerged in Chilean Antarctic stations, disrupting the summer campaign with ongoing isolation needs. The SARS-CoV-2 RNA was detected in the wastewater of several scientific stations. In Antarctica, treated wastewater is discharged directly into the seawater. No studies currently address the recovery of infectious virus particles from treated wastewater, but their presence raises the risk of infecting wildlife and initiating new replication cycles. This study highlights the initial virus detection in wastewater from Antarctic stations, identifying viral RNA via RT-qPCR targeting various genomic regions. The virus’s RNA was found in effluent from two wastewater plants at Maxwell Bay and O’Higgins Station on King George Island and the Antarctic Peninsula, respectively. This study explores the potential for the reverse zoonotic transmission of SARS-CoV-2 from humans to Antarctic wildlife due to the direct release of viral particles into seawater. The implications of such transmission underscore the need for continued vigilance and research.

## 1. Introduction

Antarctica has long been considered a geographically isolated continent [1]. Consequently, the introduction of any organism has the potential to disrupt the existing biota [2,3,4]. Even microorganisms responsible for causing infections in humans face substantial barriers in reaching the continent [5]. This is primarily attributed to the stringent regulations imposed by the signatory governments of the Antarctic Treaty, which strictly control access to Antarctica. Moreover, during the winter months, human presence is notably limited with around 90% of activities being concentrated in the spring and summer seasons. One of the advantages of maintaining controlled access to Antarctica is the preservation of a complete absence of SARS-CoV-2, the causative agent of COVID-19, on the continent for approximately nine months. However, in December 2020, the first case of COVID-19 was detected at the Chilean O’Higgins station. One of the primary concerns regarding the spread of COVID-19 in Antarctica was the limited availability of medical equipment at the research stations and the inability to transfer potentially critically ill patients to more advanced healthcare facilities. Fortunately, there were no reports of severely ill patients, and no one had to be transported to the South American continent. Nonetheless, a significant concern revolved around the potential transmission of the virus to wildlife [6,7].

The spread of the pandemic in Antarctica was closely linked to the start of summer campaigns at different research stations. As of 16 December 2020, Chilean health authorities had reported 60 confirmed cases. One of the initial steps taken to mitigate the rise in cases was to administer diagnostic tests to all personnel both at the scientific stations and before their arrival on the Antarctic continent. Nonetheless, even with infected individuals under control, it remained essential to determine whether the virus was being released into the environment. To achieve this, wastewater monitoring was implemented in some of the Chilean scientific stations. The detection of SARS-CoV-2 in this type of sample has been extensively documented in the scientific literature [8,9,10,11]. Currently, measuring and quantifying the SARS-CoV-2 in wastewater has been established as an effective surveillance measure [12,13]. Once the virus enters wastewater due to its release in the feces and secretions of infected individuals, its persistence in water is limited, as it undergoes accelerated degradation processes, especially in the marine environment [14,15,16]. This characteristic minimizes the risk of infecting potential hosts in the marine ecosystem, which could serve as animal reservoirs for the virus.

Despite the limited amount of research describing the presence of SARS-CoV-2 in marine animals, its detection has been documented in species such as Pacific oysters and other bivalves [17,18]. These filter feeders reveal the virus’s ability to reach coastal and estuarine environments. A primary concern lies in the compounds released by wastewater treatment plants (WWTPs) that utilize biological treatment systems. Several studies detail different methods through which the virus can be released, including its complete form (comprising the envelope, nucleocapsid, and genetic material), only the nucleocapsid in combination with the genetic material, or solely the genetic material [19,20]. In Antarctica, most scientific stations, whether permanent or seasonal, are equipped with biological treatment systems in their wastewater facilities [21]. Consequently, there is a potential impact on wildlife due to the release of materials contaminated with SARS-CoV-2 in this region. 

To assess the potential presence of SARS-CoV-2 traces in the WWTP of scientific stations where cases of SARS-CoV-2 infection have been identified, samples were collected from both the effluent and the influent. Furthermore, an investigation was conducted to ascertain the presence of the virus in the feces of animals inhabiting nearby areas, aiming to determine its existence in the surrounding wildlife.

## 2. Materials and Methods

### 2.1. Sampling 

Wastewater treatment plants from three Chilean scientific stations were selected to be sampled: (i) Professor Julio Escudero; (ii) President Eduardo Frei Montalva; and (iii) General Bernardo O’Higgins (Figure 1). These stations were selected based on the observation of multiple cases of COVID-19 diagnosed during the 2020–2021 summer season. The sampling consisted of 1 L of wastewater, which was collected weekly from both the influent and the effluent in each of the WWTPs between December 2020 and February–March 2021. The samples were stored at 4 °C until they were processed in the laboratory at the Universidad Andrés Bello (Santiago, Chile). The maximum arrival time of the samples at the laboratory was 10 days. Several publications have shown that the maximum time in which the samples maintain their integrity at 4 °C is 25–30 days; therefore, the samples were maintained at the ideal conditions for the SARS-CoV-2 detection analysis [22].

### 2.2. Virus Concentration 

The flocculation with skimmed milk was used to concentrate the virus following the protocols published by Calgua et al. (2013), Guerrero-Latorre et al. (2020), Melgaço et al. (2018) [23,24,25] with some modifications. Briefly, 500 mL of wastewater was pH adjusted (3.5) with 1N HCl, and then 5 mL of 1% *w*/*v* pre-flocculated skimmed milk in artificial sea water was added. This mixture was kept stirring for 8 h at room temperature. Subsequently, the sample was centrifuged at 4000× *g* for 40 min at 4 °C. Finally, the obtained pellet was resuspended in 4 mL 1× phosphate buffer (PBS) and stored at −80 °C until use.

### 2.3. RNA Extraction and Virus Detection

The extraction method used consists of 2 steps. The first is the resuspension of the pellet obtained in TRIZOL (Thermo-Fisher, Waltham, MA, USA), following the protocol of Rio et al. (2010) [26] until the aqueous phase is obtained. Subsequently, to extract the RNA from the aqueous phase, the purification kit E.Z.N.A Nucleic Acid Purification System (Omega, Knoxville, TN, USA) was used. Finally, virus detection was performed by loading between 3 and 5 L of the RNA extract, amplifying the N1, N2, E, and RdRp targets with the Taqman^®^ Fast Viral 1-Step Master Mix (ThermoFisher Scientific, Waltham, MA, USA). The gene that codes for human RNAseP was used as quality control of the genetic material. The primer sequences are compiled in Appendix A. A sample is considered positive when it amplifies positively for N1, RNAseP and any of the other targets used for virus amplification and whose Ct is less than 40. All samples were analyzed in triplicate. Those yielding negative results underwent a second round of analysis. In this phase, dilutions of 10^−1^, 10^−2^, and 10^−3^ were applied to eliminate the possibility of PCR reaction inhibition.

### 2.4. Estimation of SARS-CoV-2 Genome Copy Number in Wastewater

An absolute quantification of the genetic material was performed to determine the number of SARS-CoV-2 genomic copies in the wastewater. Calibration curves were prepared based on individual plasmids that contained copies of the SARS-CoV-2 N-code gene target. Plasmids were purchased from IDT DNA Technologies. For the targets based on the N1, the control plasmid 2019-NCoV_N_Positive was used. Using this plasmid, calibration curves were constructed over a range of copy numbers, spanning from 1 × 10^6^ up to 1 × 10^0^ copies (1,000,000 to 1 copies). The number of genomic copies was obtained through the Ct value of each of the samples in each qRT-PCR reaction, which is interpolated in the equation of the line obtained with the calibration curves. The obtained value corresponds to the number of genomic equivalents copies of the SARS-CoV-2 virus per liter of wastewater [27]. As a consensus from multiple studies, viral load is reported based on the number of genomic equivalents obtained using the N1 target [24,28,29,30,31]. The sensitivity of the technique was assessed through serial dilutions of the calibration curve for each of the genes analyzed. It was determined that the technique could detect as little as one copy per liter (copy/L) of the N1 and N2 targets and 10 copies/L of the E and RdRp genes, as described by Olivares-Pacheco et al. [10] A comparable method was employed to assess the specificity of the primers utilized [10]. To mitigate the risk of false positives, all samples identified as positive underwent Sanger sequencing of the N1 amplicon.

### 2.5. Genome Sequencing and Variant Typing of SARS-CoV-2

The COVIDSeq commercial platform on an Illumina NextSeq500 machine with the V4 primer pool was used to genotype SARS-CoV-2 positive samples. Briefly, the protocol amplifies 98 viral targets based on the ARTIC V3 protocol (https://artic.network/ncov-2019, accessed on 31 January 2024). The presence of SARS-CoV-2 in the samples was established with a threshold of >90 targets detected per sample using default thresholds. Then, variant typing, both Pangolin and NextClade systems, was conducted using the DRAGEN COVID lineage app, which is freely available at Illumina BaseSpace (https://basespace.illumina.com/) accessed on 31 July 2023.

### 2.6. Environmental Samples and SARS-CoV-2 Detection in Antarctic Wildlife

Samples were collected during the Chilean Antarctic Expedition 58 (ECA58). ECA58 took place from 13 January to 9 February 2022 in the Antarctic Peninsula Isabel Riquelme Islet (63°19′5″ S and 57°53′55″ W). Sampling during ECA58 was performed at the General Bernardo O’Higgins station penguin colony site. A total of 105 direct environmental samples from Antarctic fauna were collected using sterile swabs and placed in 1.5 mL microtubes containing viral transport medium (VTM) from VQIR, catalog number 611901 [32]. The samples were kept at ambient temperature in a cooler (nearing 0 °C) before being finally stored at −80 °C. A total of 105 samples were collected, comprising environmental samples from snowy sheathbill (fecal samples) (*Chionis albus*; *n* = 31); Antarctic tern (*Sterna vittata*; *n* = 33); chinstrap penguin (*Pygoscelis antarctica*; *n* = 26); gentoo penguin (*Pygoscelis papua*; *n* = 14); Antarctic fur seal (*Arctocephalus gazaella*; *n* = 3); Wedell seal (*Leptonychotes weddelli*; *n* = 1); and Kelp gull (*Larus dominicanus*; *n* = 4). These samples were grouped into pools of 5 samples each, resulting in a total of 21 pools. All samples were stored at −80 °C. Additionally, all sampled individuals underwent veterinary examination. 

### 2.7. RNA Extraction from Wildlife Samples

The samples were homogenized in Viral Transport Media (VTM). After decanting the particles present in the sample, 60 µL of supernatant was taken, and pools were prepared for every 5 samples with 100 µL of VTM. The pools were clarified by subsequent centrifugation at 8000× *g* for 10 min (min) at 4 °C and stored at −80 °C. RNA was extracted from 150 µL of supernatant obtained in the previous step, with TRIzol^®^ Reagent (INVITROGEN, Waltham, MA, USA) based on the Chomczynski and Sacchi (1987) [33] method, following the manufacturer’s instructions. The RNA obtained was resuspended in 30 µL of nuclease-free water and stored at −20 °C. The RNA concentration was analyzed by absorption using a SYNERGY HTX multimodal reader (Agilent Technologies, Santa Clara, CA, USA).

### 2.8. RT-qPCR and RT-PCR Analysis

PCR screening for coronavirus, paramixovirus, and influenza virus: RT-PCR was performed with Brilliant III Ultra-Fast qRT-PCR Master Mix (Agilent Technologies, USA) [34]. Coronavirus screening was performed using pancoronavirus one-step RT-PCR based on degenerate primers for a conserved 180 bp region in the polymerase gene (IZS-FW 5′-CDCAYGARTTYTGYTCNCARC-3′; IZS-RV 5′-RHGGRTANGCRTCWATDGC-3′). RT-PCR was performed at 50 °C for 30 min, which was followed by DNA polymerase activation at 95 °C for 3 min, and for 50 cycles in three steps: 95 °C for 15 s, 45 °C for 30 s, 60 °C for 15 s and a final extension at 60 °C for 60 min. The PCR product was run on a 2% agarose gel in 1× TAE at a constant 80 V for 40 min. Paramyxovirus screening was performed by a one-step panparamyxovirus RT-PCR based on primers for a conserved 121 bp region in domain III of the RNA-dependent RNA polymerase gene (PMX1-FW 5′-GARGGIYIITGYCARAARNTNTGGAC-3′; PMX2-RV 5′-TIAYIGCWATIRIYTGRTTRTCNCC-3′) [35]. Influenza virus screening RT-PCR was performed at 50 °C for 30 min, followed by DNA polymerase activation at 95 °C for 3 min, and for 50 cycles in three steps: 95 °C for 15 s, 41 °C for 30 s, 60 °C for 15 s and a final extension at 60 °C for 60 min (InfAFW-FW 5′-GACCRATCCTGTCACCTCTGAC-3′; InfAR 5′-AGGGCATTYTGGACAAAKCGTCTA-3′, InfAP 5′-FAM-TGCAGTCCTCGCTCACTGGGCACG-BHQ1-3) [32]. 

For SARS-CoV-2 RNA detection, we employed the TaqMan 2019-nCoV Assay Kit v1 (ThermoFisher Scientific, USA). This kit contains a set of TaqMan RT-PCR assays for the detection of SARS-CoV-2 RNA and includes three assays targeting the SARS-CoV-2 genes (ORF1ab, S, and N) and one control assay for the human RNase P gene.

## 3. Results

### 3.1. SARS-CoV-2 RNA Detection in WWTPs

The presence of SARS-CoV-2 was determined using qPCR assays, evaluating the N1, N2, E, and RdRp targets. A sample was considered positive if it showed amplification in N1 and any of the other targets measured. The viral load was determined based on the N1 value and is expressed as the number of genomic copies per liter. Out of the 20 samples analyzed, 12 were classified as positive (60%). The first positive sample was detected at the Escudero station on 21 December 2020, coinciding with the symptomatic and asymptomatic cases confirmed by the Chilean health authorities. This sample had a viral load of 121,600 genomic copies per liter in the influent, while 14,670 genomic copies per liter were detected in the effluent. The latter represents a significant amount of genetic material being directly released into the Fildes Bay (Table 1). In this database, the presence of the virus was also detected on 17 and 25 February 2021, but only in the tributary, with values not exceeding 2000 genomic copies per liter (Table 1). This could be explained as a residue of what occurred in December and January, since no positive cases were detected in February. 

Regarding the President Frei station, which is geographically close to the Escudero station, the virus was detected in its respective WWTP only on 10 February but in very low quantities (less than 2000 copies per liter in the tributary and less than 1000 copies in the effluent). No cases of infected individuals were reported on that date; hence, it can be attributed to the presence of asymptomatic cases. The absence of symptoms implies a lack of testing on individuals.

A different case occurred at the O’Higgins station. This station consistently showed the presence of the virus during the sampling conducted on 23 February, 2 March, and 9 March 2021 (Table 1). The highest viral load was recorded on 9 March with 41,000 copies per liter in the influent and 3700 copies per liter in the effluent. The second highest concentration was observed in the sample collected on 23 February with an influent load of 26,000 copies per liter and an effluent load of 5300 copies per liter. Lastly, on 2 March, the influent load was 9700 copies, while the effluent load was 7100 copies per liter (Table 1). These figures clearly indicate a significant amount of genetic material being released into the environment. It is important to note that none of the samples collected on these dates confirmed any cases of infection among individuals. This highlights the possibility of “silent outbreaks” occurring more frequently than reported in enclosed and isolated facilities, such as the Antarctic bases.

We also attempted to recover full SARS-CoV-2 genomes from samples coming from wastewater and the research stations. We were able to successfully amplify and sequence 9 samples, all of which had Ct values of less than 35. In accordance with circulating lineages in Chile at the time of sampling, all the sequence genomes belonged to the B1 lineage, including two subvariants: B.1.1.451 and B.1.1.409 (Table 1).

### 3.2. Environmental Animal Viral Detections by Real-Time RT-PCR Assays

The samples were examined for the presence of virus using TaqMan-based real-time reverse transcriptase-PCR (rRT-PCR) targeting the matrix gene. Virus screening from environmental samples included SARS-CoV-2, coronavirus, paramyxovirus, and influenza virus. These virus genes were detected in 13 out of 21 pool samples (61.90%). Pan-coronavirus and pan-paramixovirus was confirmed in 6 and 12 of 21 environmental pool tested, respectively. These same samples (21 pool) were negative for SARS-CoV-2 and influenza virus identification (Table 2). While a few pools included mammal samples such as Antarctic fur seal (*Arctocephalus gazaella*) or Wedell seal (*Leptonychotes weddelli*) (Pools 1 and 2), the rest included Antarctic birds such as gulls, snowy sheathbill, and penguins. The snowy sheathbill presented a high positivity for paramyxovirus and pan-coronavirus (no SARS-CoV-2); however, the high positivity could be associated with endemic viral microbiota plus its scavenger behavior (Figure 2).

## 4. Discussion

The first COVID-19 outbreak in Antarctica was reported in December 2020 with 60 positive cases confirmed by PCR test. These cases were detected at scientific stations in the South Shetland Islands and the Antarctic Peninsula, raising concerns about the potential contact between infected humans and wildlife, as well as the release of fecal material into the marine environment. Hughes and Convey (2020) [7] expressed their concern regarding the potential for zoonotic transmission of SARS-CoV-2 from humans to Antarctic wildlife, which could lead to rapid spread within colonies and even mass mortality events among animals or the creation of a new reservoir with zoonotic potential [36,37]. In this regard, multiple human-to-animal spillover events of SARS-CoV-2 have been reported [38,39,40], evidencing the plasticity of the virus to infect a large range of hosts. The virus has been confirmed in domestic, peri-domestic, and wildlife populations. Indeed, SARS-CoV-2 has been able to establish and adapt to whitetail deer populations in North America [41]. Therefore, the human-to-animal spillover is a fact and could be possible for Antarctic wildlife. Moreover, in silico analysis of ACE2 indicates the possibility of SARS-CoV-2 to spillover to still not confirmed species including mammalian and avian [42], suggesting that Antarctic wildlife, especially marine mammals, could be susceptible to the virus [6,43]. 

Our data indicate the presence of SARS-CoV-2 in the WWTPs of the scientific Antarctic stations, each with variations in their technology. While the WWTP at Escudero and O’Higgins stations utilize both mechanical and biological processes, the Frei WWTP employs a red worm in the treatment process, which feed on bacteria in both the activated sludge and trickling filter systems. Ultimately, UV light is employed for purification at all three stations as part of the final treatment phase. Several authors have previously reported the potential release of the virus into the environment. Randazzo et al. (2020) [27] demonstrated that WWTPs with only secondary treatment can release SARS-CoV-2 RNA in their effluents, mirroring the findings of our study in the WWTPs sampled [44], suggesting that at least part of SARS-CoV-2 virions or their RNA present in sewage effluents may flow into watercourses and eventually be released into coastal areas. 

Interestingly, the initial detection of SARS-CoV-2 at King George Island aligns with the first outbreak reported in December 2020. The substantial number of genomic copies identified at the Escudero WWTP was substantiated by the isolation of a single symptomatic suspect case, following the protocol established for suspected cases in Antarctica. However, the continued presence of SARS-CoV-2 in the WWTP could be attributed to asymptomatic cases that went undetected during quarantine due to certain shortcomings in the protocol. This is even though all personnel were tested before their arrival in Antarctica. SARS-CoV-2 is a recent emergence, and we require more data to comprehend its behavior in the environment. An analysis of surrogate coronaviruses survivability in water and sewage was conducted by Casanova et al. (2009) [45], revealing their persistence and infectiousness at both low (4 °C) and moderate (25 °C) temperatures [46]. Additionally, some studies have reported that certain coronaviruses can endure extreme cold, surviving for years at temperatures as low as −60 °C while retaining their infectious properties [47]. These lower temperatures might contribute to the viral particles’ persistence in polar environments. 

Other factors, such as organic matter or solid fraction in water, could enhance the survival of the viral population. Organic matter particles, for instance, can physically shield the virus, as documented by Paul et al. (2021) [46]. The detection of SARS-CoV-2 RNA in the influent wastewater at the Antarctic station suggests that WWTPs, influenced by factors like temperature and organic matter, might intensify the potential interaction of the virus with wildlife. This necessitates the quantification of the infective dose, determination of the number of viable virus particles in feces, and the collection of additional data regarding its viability in water systems [48]. 

However, it is essential to note that the presence of SARS-CoV-2 RNA in the aquatic environment does not necessarily confirm the presence of infectious virus. Estimating the number of viable virus copies requires knowing the proportion of infectious virus in wastewater [49]. Consequently, further research is necessary for WWTPs in Antarctica. Based on the current literature and scientific evidence to date, it is crucial enhance the efficiency of graywater treatment plant systems, potentially incorporating new technologies [50]. One consideration could be the addition of a final disinfection step, like ozonation of wastewater, to further reduce the risk posed by viral pathogens, such as SARS-CoV-2, before discharge into the sea. Also, it is equally important to ensure the timely replacement and optimal maintenance of UV lamps if they are used in the final treatment process [46].

Coronaviruses have been identified from a variety of wild birds and mammals [51,52]. Among wild birds, gammacoronavirus is the predominant type of CoV, which is followed by deltacoronavirus [53]. Notably, a potential species of deltacoronavirus has been observed in healthy Antarctic penguins [54,55]. Although this virus does not induce illness, its presence in these penguins underscores the extensive geographical coronaviruses of this type. 

The initial evidence of SARS-CoV-2 presence emerged in clam populations of the *Ruditapes* sp. genus due to untreated water discharges in Galicia, Spain [44]. A related species, the bivalve filter-feeder known as the Antarctic clam (*Laternula elliptica*), has the potential to accumulate viral RNA in areas near WWTPs discharges. Aquatic mammals such as cetaceans, including the Antarctic minke whale (*Balaenoptera bonaerensis*) and killer whale (*Orcinus orca*), as well as the Antarctic fur seals and Wedell seals, which retain many key receptor binding domains for SARS-CoV-2, are subjects for assessing the virus persistence [56]. Further investigations are required to evaluate the susceptibility of Antarctic mammals to coronaviruses, including SARS-CoV-2 [6]. However, our study did not detect SARS-CoV-2-RNA-positive free-ranging animals, suggesting that there was no widespread circulation among the few species examined during the study period. 

Invasive species such as insects on King George Island generate a significant concern, and this concern should be heightened in the current pandemic context. Most of the insects arriving in Antarctica are associated with treatment plants, increasing the likelihood of interaction with contaminated feces. It is essential to continue monitoring insects like *Trichocera maculipennis* found in the treatment plants of King George Island, which could potentially serve as vectors for virus transmission to the environment through routes other than aquatic [57]. Although the ACE2 receptor in insects differs significantly from that of mammals, making efficient binding with SARS-CoV-2 unlikely, it has been reported that arthropods were involved in the mechanical transmission of the turkey coronavirus, parapoxvirus, and SARS-CoV-2 [58,59]. Experimental studies have demonstrated that houseflies may be a vector for SARS-CoV-2 genomic RNA transmission to the surrounding environment up to 24 h post-exposure [60]. Finally, it is advisable to implement a monitoring program to assess the potential presence of SARS-CoV-2 in WWTP using PCR techniques or biosensor-based technologies, which have been extensively employed for virus detection. This monitoring should also include an evaluation of the impact of human activities on the Antarctic ecosystem.

## Figures and Tables

**Figure 1 microorganisms-12-00743-f001:**
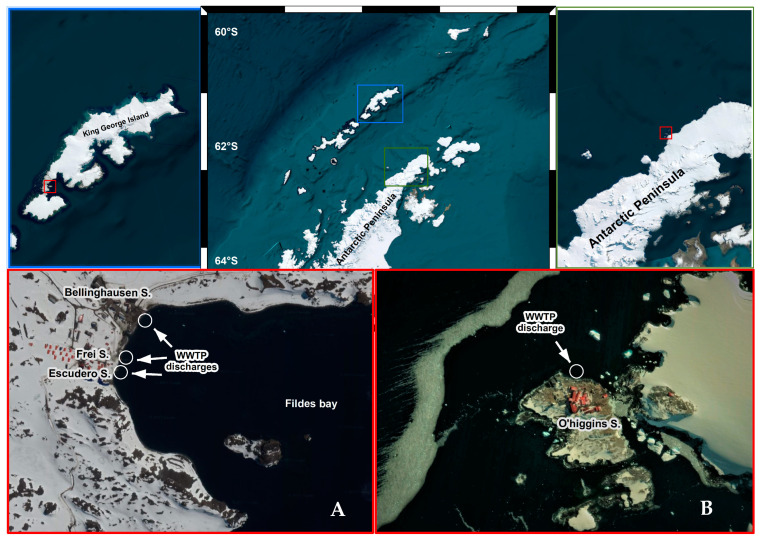
Overall view of the sampling area in King George Island and Antarctic Peninsula. Map of site locations of WWTPs in Fildes Peninsula (**A**) and General Bernardo O’Higgins station (**B**).

**Figure 2 microorganisms-12-00743-f002:**
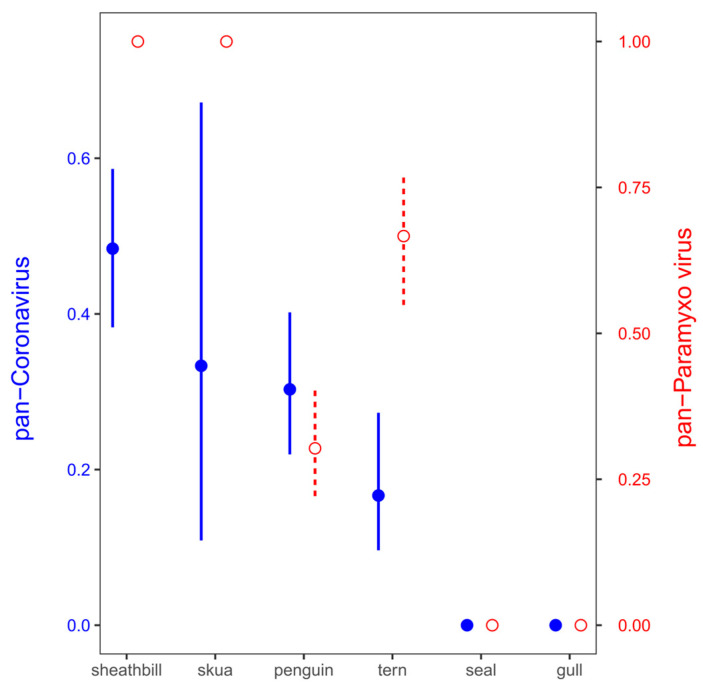
Positivity for paramyxovirus (red empty circles and dashed lines) and coronavirus (blue filled circles and solid lines) in Antarctic birds. Mean ± standard deviation probabilities of viral detection in the pools were calculated as a proportion of pools with positive detection (1) against no detection (0) from all the 22 pools.

**Table 1 microorganisms-12-00743-t001:** SARS-CoV-2 detection in the wastewater samples taken at different Chilean Antarctic stations from 21 December 2020 to 9 March 2021 (*n* = 11). I: Influent; E: Effluent; (+) positive sample; n/s: Not sampled; n/d: Not detected. No sequence: No sequence obtained.

Date	Source	Frei	Escudero	O’Higgins	Lineages Detected
21 December 2020	I	n/s	(+)	n/s	B.1.1.451
	E	n/s	(+)	n/s	B.1.1
10 February 2021	I	(+)	n/d	n/s	B.1.1
	E	(+)	n/d	n/s	B.1.1
17 February 2021	I	n/d	n/d	n/s	n/s
	E	n/d	(+)	n/s	No sequence
23 February 2021	I	n/s	n/s	(+)	B.1.1
	E	n/s	n/s	(+)	B.1.1
24 February 2021	I	n/d	n/s	n/s	n/s
	E	n/d	n/s	n/s	n/s
25 February 2021	I	n/s	n/d	n/s	n/s
	E	n/s	(+)	n/s	No sequence
2 March 2021	I	n/s	n/s	(+)	B.1.1
	E	n/s	n/s	(+)	B.1.1
3 March 2021	I	n/s	n/d	n/s	n/s
	E	n/s	n/d	n/s	n/s
9 March 2021	I	n/s	n/s	(+)	B.1.1.409
	E	n/s	n/s	(+)	No sequence

**Table 2 microorganisms-12-00743-t002:** Results of PCR testing of environmental animal samples collected around O’Higgins station for virus panel. (+) Positive sample (−) negative sample.

Environmental Pool	Pan-Coronavirus	SARS-CoV-2	Pan-Paramyxovirus	Influenza A
1	(−)	(−)	(−)	(−)
2	(−)	(−)	(−)	(−)
3	(−)	(−)	(−)	(−)
4	(−)	(−)	(−)	(−)
5	(−)	(−)	(+)	(−)
6	(+)	(−)	(+)	(−)
7	(−)	(−)	(+)	(−)
8	(−)	(−)	(+)	(−)
9	(+)	(−)	(+)	(−)
10	(−)	(−)	(−)	(−)
11	(−)	(−)	(−)	(−)
12	(+)	(−)	(−)	(−)
13	(−)	(−)	(−)	(−)
14	(−)	(−)	(−)	(−)
15	(−)	(−)	(+)	(−)
16	(−)	(−)	(+)	(−)
17	(+)	(−)	(+)	(−)
18	(+)	(−)	(+)	(−)
19	(−)	(−)	(+)	(−)
20	(+)	(−)	(+)	(−)
21	(−)	(−)	(+)	(−)
Negative control	(−)	(−)	(−)	(−)

## Data Availability

Data are contained within the article and Appendix A.

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
