# Peer review of "Detection of SARS-CoV-2 in Wastewater Associated with Scientific Stations in Antarctica and Possible Risk for Wildlife"

_microorganisms, 2024, doi:10.3390/microorganisms12040743_

Round 1
Reviewer 1 Report
Comments and Suggestions for Authors
This study has raised little serious issue of possible transfer of SARS-CoV-2 back into other organisms in the nature. Anything is possible in nature, but still viruses are highly hostspecific. Many earlier studies mentioned, detection of SARS-CoV-2 markers in wastewater does not guarantee the prevalence of pathogenic virom (https://doi.org/10.1007/s11356-021-16919-3, https://doi.org/10.1016/j.watres.2022.118220, https://doi.org/10.1093/femsmc/xtab007), even if a pathogenic strain is alive in nature, it is less likely to adopt in a new host. It is an empirical experiment, where they detected virom markers in nature. It’s fine to publish the paper, but they have to be more critical about the issue I have raised.
Author Response
We appreciate and consider the comments made by the reviewer. In light of this, the following paragraph has been included in the discussion section:
“In this regard, multiple human-to-animal spillover events of SARS-CoV-2 have been reported (Amoutzias et al., 2022; Cui et al., 2022; Tan et al., 2022), evidencing the plas-ticity of the virus to infect a large range of hosts. The virus has been confirmed in domestic, peri-domestic, and wildlife populations. Indeed, SARS-CoV-2 has been able to establish and adapt to While-Tail Deer populations in North America (Feng et al., 2023). Therefore, the human-to-animal spillover is a fact and could be possible for Antarctic wildlife. Moreover, in silico analysis of ACE2 insights the possibility of SARS-CoV-2 to spill over to still not confirmed species including mammalian and avian (Mesquita et al., 2023), suggesting that Antarctic wildlife, especially marine mammals could be susceptible to the virus (Barbosa et al., 2021; Gryseels et al., 2021).”
Amoutzias, G.D., M. Nikolaidis, E. Tryfonopoulou, K. Chlichlia, P. Markoulatos, and S.G. Oliver, 2022: The Remarkable Evolutionary Plasticity of Coronaviruses by Mutation and Recombination: Insights for the COVID-19 Pandemic and the Future Evolutionary Paths of SARS-CoV-2. Viruses 14, 78, DOI: 10.3390/v14010078.
Barbosa, A., A. Varsani, V. Morandini, W. Grimaldi, R.E.T. Vanstreels, J.I. Diaz, T. Boulinier, M. Dewar, D. González-Acuña, R. Gray, C.R. McMahon, G. Miller, M. Power, A. Gamble, and M. Wille, 2021: Risk as-sessment of SARS-CoV-2 in Antarctic wildlife. Science of The Total Environment 755, 143352, DOI: 10.1016/j.scitotenv.2020.143352.
Cui, S., Y. Liu, J. Zhao, X. Peng, G. Lu, W. Shi, Y. Pan, D. Zhang, P. Yang, and Q. Wang, 2022: An Updated Review on SARS-CoV-2 Infection in Animals. Viruses 14, 1527, DOI: 10.3390/v14071527.
Feng, A., S. Bevins, J. Chandler, T.J. DeLiberto, R. Ghai, K. Lantz, J. Lenoch, A. Retchless, S. Shriner, C.Y. Tang, S.S. Tong, M. Torchetti, A. Uehara, and X.-F. Wan, 2023: Transmission of SARS-CoV-2 in free-ranging white-tailed deer in the United States. Nat Commun 14, 4078, DOI: 10.1038/s41467-023-39782-x.
Gryseels, S., L. De Bruyn, R. Gyselings, S. Calvignac‐Spencer, F.H. Leendertz, and H. Leirs, 2021: Risk of human‐to‐wildlife transmission of SARS‐CoV‐2. Mamm Rev 51, 272–292, DOI: 10.1111/mam.12225.
Mesquita, F.P., P.F. Noronha Souza, D.R. Aragão, E.M. Diógenes, E.L. da Silva, J.L. Amaral, V.N. Freire, D. de Souza Collares Maia Castelo-Branco, and R.C. Montenegro n.d.: In silico analysis of ACE2 from different animal species provides new insights into SARS-CoV-2 species spillover. Future Virol10.2217/fvl-2022–0187, DOI: 10.2217/fvl-2022-0187.
Tan, C.C.S., S.D. Lam, D. Richard, C.J. Owen, D. Berchtold, C. Orengo, M.S. Nair, S.V. Kuchipudi, V. Kapur, L. van Dorp, and F. Balloux, 2022: Transmission of SARS-CoV-2 from humans to animals and potential host adaptation. Nat Commun 13, 2988, DOI: 10.1038/s41467-022-30698-6.
Reviewer 2 Report
Comments and Suggestions for Authors
The COVID-19 pandemic has posed challenges for the remote and isolated continent of Antarctica. The detection of SARS-CoV-2 RNA in Antarctic wastewater raises concerns about the potential for reverse zoonotic transmission from humans to Antarctic wildlife. While no studies have yet addressed the recovery of infectious virus particles from treated wastewater, the presence of viral RNA suggests a risk of infecting wildlife and initiating new replication cycles. This highlights the importance of continued vigilance and research to understand the implications of such transmission in the Antarctic ecosystem. This study provides initial evidence of the presence of SARS-CoV-2 RNA in wastewater from Antarctic stations, indicating the potential for the release of viral particles into the surrounding seawater. The implications of reverse zoonotic transmission to Antarctic wildlife underscore the need for ongoing monitoring and research to mitigate potential risks and protect the unique ecosystem of Antarctica. I recommend accepting this article after MINOR REVISIONS.
1. There is a lack of related literature citations. Please include relevant content in the relate sections.
2. It would be beneficial to add the shortcomings of current studies and areas for improvement. It is advisable to add some personal reflections and discussions on the strengths, limitations, and future directions of the article.
3. In “Discussion” section, the quality must be improved. It is better to provide more solid evidence and strengthen the validity to explore their relevance.
Comments on the Quality of English LanguageMinor editing of English language
Author Response
We appreciate the reviewer's comments and accept them.
In response to point “There is a lack of related literature citations. Please include relevant content in the relate sections”, we have included current and relevant literature pertaining to the topic under discussion.
Responding to point “It would be beneficial to add the shortcomings of current studies and areas for improvement. It is advisable to add some personal reflections and discussions on the strengths, limitations, and future directions of the article”. Personal reflections have been incorporated into the discussion of the article.
Both aspects are addressed in the following paragraph.
“In this regard, multiple human-to-animal spillover events of SARS-CoV-2 have been reported (Amoutzias et al., 2022; Cui et al., 2022; Tan et al., 2022), evidencing the plas-ticity of the virus to infect a large range of hosts. The virus has been confirmed in domestic, peri-domestic, and wildlife populations. Indeed, SARS-CoV-2 has been able to establish and adapt to While-Tail Deer populations in North America (Feng et al., 2023). Therefore, the human-to-animal spillover is a fact and could be possible for Antarctic wildlife. Moreover, in silico analysis of ACE2 insights the possibility of SARS-CoV-2 to spill over to still not confirmed species including mammalian and avian (Mesquita et al., 2023), suggesting that Antarctic wildlife, especially marine mammals could be susceptible to the virus (Barbosa et al., 2021; Gryseels et al., 2021).”
Amoutzias, G.D., M. Nikolaidis, E. Tryfonopoulou, K. Chlichlia, P. Markoulatos, and S.G. Oliver, 2022: The Remarkable Evolutionary Plasticity of Coronaviruses by Mutation and Recombination: Insights for the COVID-19 Pandemic and the Future Evolutionary Paths of SARS-CoV-2. Viruses 14, 78, DOI: 10.3390/v14010078.
Barbosa, A., A. Varsani, V. Morandini, W. Grimaldi, R.E.T. Vanstreels, J.I. Diaz, T. Boulinier, M. Dewar, D. González-Acuña, R. Gray, C.R. McMahon, G. Miller, M. Power, A. Gamble, and M. Wille, 2021: Risk as-sessment of SARS-CoV-2 in Antarctic wildlife. Science of The Total Environment 755, 143352, DOI: 10.1016/j.scitotenv.2020.143352.
Cui, S., Y. Liu, J. Zhao, X. Peng, G. Lu, W. Shi, Y. Pan, D. Zhang, P. Yang, and Q. Wang, 2022: An Updated Review on SARS-CoV-2 Infection in Animals. Viruses 14, 1527, DOI: 10.3390/v14071527.
Feng, A., S. Bevins, J. Chandler, T.J. DeLiberto, R. Ghai, K. Lantz, J. Lenoch, A. Retchless, S. Shriner, C.Y. Tang, S.S. Tong, M. Torchetti, A. Uehara, and X.-F. Wan, 2023: Transmission of SARS-CoV-2 in free-ranging white-tailed deer in the United States. Nat Commun 14, 4078, DOI: 10.1038/s41467-023-39782-x.
Gryseels, S., L. De Bruyn, R. Gyselings, S. Calvignac‐Spencer, F.H. Leendertz, and H. Leirs, 2021: Risk of human‐to‐wildlife transmission of SARS‐CoV‐2. Mamm Rev 51, 272–292, DOI: 10.1111/mam.12225.
Mesquita, F.P., P.F. Noronha Souza, D.R. Aragão, E.M. Diógenes, E.L. da Silva, J.L. Amaral, V.N. Freire, D. de Souza Collares Maia Castelo-Branco, and R.C. Montenegro n.d.: In silico analysis of ACE2 from different animal species provides new insights into SARS-CoV-2 species spillover. Future Virol10.2217/fvl-2022–0187, DOI: 10.2217/fvl-2022-0187.
Tan, C.C.S., S.D. Lam, D. Richard, C.J. Owen, D. Berchtold, C. Orengo, M.S. Nair, S.V. Kuchipudi, V. Kapur, L. van Dorp, and F. Balloux, 2022: Transmission of SARS-CoV-2 from humans to animals and potential host adaptation. Nat Commun 13, 2988, DOI: 10.1038/s41467-022-30698
Reviewer 3 Report
Comments and Suggestions for Authors
The following comments may be helpful to improve the manuscript:
1. The manuscript describes the use of RT-qPCR for detecting SARS-CoV-2 in wastewater samples. Could the authors provide more details on the sensitivity and specificity of this method in the context of wastewater analysis? Additionally, how were false positives or negatives accounted for in your study?
2. The study discusses the potential risk of SARS-CoV-2 transmission to Antarctic wildlife. However, there seems to be a lack of detailed analysis or discussion on the mechanisms of transmission and the actual risk levels. Could the authors elaborate on how the virus might affect different species in the Antarctic ecosystem, and whether there are any preventive measures that can be taken?
3. In the results section, the authors mention the detection of SARS-CoV-2 RNA in wastewater but also highlight the absence of SARS-CoV-2 RNA in the environmental samples from Antarctic wildlife. Could the authors discuss how this finding impacts the overall conclusions of your study, especially in relation to the potential threat of SARS-CoV-2 to Antarctic wildlife?
Author Response
We appreciate the reviewer's comments and have addressed their concerns accordingly.
In response to comment “The manuscript describes the use of RT-qPCR for detecting SARS-CoV-2 in wastewater samples. Could the authors provide more details on the sensitivity and specificity of this method in the context of wastewater analysis? Additionally, how were false positives or negatives accounted for in your study?”, strategies aimed at enhancing the specificity and sensitivity of the SARS-CoV-2 detection technique in wastewater have been incorporated into the manuscript. Additionally, this revised version of the manuscript discusses the measures taken to address both false negatives and false positives.
Replying to comments “The study discusses the potential risk of SARS-CoV-2 transmission to Antarctic wildlife. However, there seems to be a lack of detailed analysis or discussion on the mechanisms of transmission and the actual risk levels. Could the authors elaborate on how the virus might affect different species in the Antarctic ecosystem, and whether there are any preventive measures that can be taken?” and “In the results section, the authors mention the detection of SARS-CoV-2 RNA in wastewater but also highlight the absence of SARS-CoV-2 RNA in the environmental samples from Antarctic wildlife. Could the authors discuss how this finding impacts the overall conclusions of your study, especially in relation to the potential threat of SARS-CoV-2 to Antarctic wildlife?”. This is addressed in the following paragraph included in the manuscript
“In this regard, multiple human-to-animal spillover events of SARS-CoV-2 have been reported (Amoutzias et al., 2022; Cui et al., 2022; Tan et al., 2022), evidencing the plas-ticity of the virus to infect a large range of hosts. The virus has been confirmed in domestic, peri-domestic, and wildlife populations. Indeed, SARS-CoV-2 has been able to establish and adapt to While-Tail Deer populations in North America (Feng et al., 2023). Therefore, the human-to-animal spillover is a fact and could be possible for Antarctic wildlife. Moreover, in silico analysis of ACE2 insights the possibility of SARS-CoV-2 to spill over to still not confirmed species including mammalian and avian (Mesquita et al., 2023), suggesting that Antarctic wildlife, especially marine mammals could be susceptible to the virus (Barbosa et al., 2021; Gryseels et al., 2021).”
Amoutzias, G.D., M. Nikolaidis, E. Tryfonopoulou, K. Chlichlia, P. Markoulatos, and S.G. Oliver, 2022: The Remarkable Evolutionary Plasticity of Coronaviruses by Mutation and Recombination: Insights for the COVID-19 Pandemic and the Future Evolutionary Paths of SARS-CoV-2. Viruses 14, 78, DOI: 10.3390/v14010078.
Barbosa, A., A. Varsani, V. Morandini, W. Grimaldi, R.E.T. Vanstreels, J.I. Diaz, T. Boulinier, M. Dewar, D. González-Acuña, R. Gray, C.R. McMahon, G. Miller, M. Power, A. Gamble, and M. Wille, 2021: Risk as-sessment of SARS-CoV-2 in Antarctic wildlife. Science of The Total Environment 755, 143352, DOI: 10.1016/j.scitotenv.2020.143352.
Cui, S., Y. Liu, J. Zhao, X. Peng, G. Lu, W. Shi, Y. Pan, D. Zhang, P. Yang, and Q. Wang, 2022: An Updated Review on SARS-CoV-2 Infection in Animals. Viruses 14, 1527, DOI: 10.3390/v14071527.
Feng, A., S. Bevins, J. Chandler, T.J. DeLiberto, R. Ghai, K. Lantz, J. Lenoch, A. Retchless, S. Shriner, C.Y. Tang, S.S. Tong, M. Torchetti, A. Uehara, and X.-F. Wan, 2023: Transmission of SARS-CoV-2 in free-ranging white-tailed deer in the United States. Nat Commun 14, 4078, DOI: 10.1038/s41467-023-39782-x.
Gryseels, S., L. De Bruyn, R. Gyselings, S. Calvignac‐Spencer, F.H. Leendertz, and H. Leirs, 2021: Risk of human‐to‐wildlife transmission of SARS‐CoV‐2. Mamm Rev 51, 272–292, DOI: 10.1111/mam.12225.
Mesquita, F.P., P.F. Noronha Souza, D.R. Aragão, E.M. Diógenes, E.L. da Silva, J.L. Amaral, V.N. Freire, D. de Souza Collares Maia Castelo-Branco, and R.C. Montenegro n.d.: In silico analysis of ACE2 from different animal species provides new insights into SARS-CoV-2 species spillover. Future Virol10.2217/fvl-2022–0187, DOI: 10.2217/fvl-2022-0187.
Tan, C.C.S., S.D. Lam, D. Richard, C.J. Owen, D. Berchtold, C. Orengo, M.S. Nair, S.V. Kuchipudi, V. Kapur, L. van Dorp, and F. Balloux, 2022: Transmission of SARS-CoV-2 from humans to animals and potential host adaptation. Nat Commun 13, 2988, DOI: 10.1038/s41467-022-30698-6.
Reviewer 4 Report
Comments and Suggestions for Authors
The submitted manuscript deals with the detection of SARS-CoV-2 genetic material in the wastewater of several polar stations and in environmental material. Although the results are not unexpected, the work has an undoubted scientific contribution. Overall, the manuscript is of high quality both in terms of methodology and interpretation, it is written clearly and legibly.
Minor notes:
Figures 1 and 2 must be supplied in a better quality/more suitable format.
There are inconsistencies in line spacing in several places, they must be corrected (lines 138, 146, 161 and 194)
Author Response
Thank you very much for the comments raised by the reviewer. In response to your observations, we have included improved versions of the images and corrected the spacing on lines 138, 146, 161, and 194.